# Facile Synthesis of Unsupported Pd Aerogel for High Performance Formic Acid Microfluidic Fuel Cell

**DOI:** 10.3390/ma15041422

**Published:** 2022-02-15

**Authors:** Alejandra Martínez-Lázaro, Luis A. Ramírez-Montoya, Janet Ledesma-García, Miguel A. Montes-Morán, Mayra P. Gurrola, J. Angel Menéndez, Ana Arenillas, Luis G. Arriaga

**Affiliations:** 1División de Investigación y Posgrado, Facultad de Ingeniería, Universidad Autónoma de Querétaro, Santiago de Queretaro 76010, Mexico; AleeM.Lazaro@live.com (A.M.-L.); janet.ledesma@uaq.mx (J.L.-G.); 2Laboratory for Research on Advanced Processes for Water Treatment, Engineering Institute, Universidad Nacional Autónoma de México (UNAM), Blvd. Juriquilla 3001, Santiago de Queretaro 76230, Mexico; lar-75@hotmail.com; 3Instituto de Ciencia y Tecnología del Carbono, INCAR-CSIC. Francisco Pintado Fe, 33011 Oviedo 26., Spain; miguel@incar.csic.es (M.A.M.-M.); angelmd@incar.csic.es (J.A.M.); 4CONACYT-Tecnológico Nacional de México/Instituto Tecnológico de Chetumal. Av. Insurgentes 330, David Gustavo Gutiérrez, Chetumal 77013, Mexico; mayra.pg@chetumal.tecnm.mx; 5Centro de Investigación y Desarrollo Tecnológico en Electroquímica, Santiago de Queretaro 76703, Mexico

**Keywords:** unsupported aerogel, microwave heating, microfluidic fuel cell, electro-oxidation

## Abstract

In this work, unsupported Pd aerogel catalysts were synthesized for the very first time by using microwaves as a heating source followed by a lyophilization drying process and used towards formic acid electro-oxidation in a microfluidic fuel cell. Aerogels were also made by heating in a conventional oven to evaluate the microwave effect during the synthesis process of the unsupported Pd aerogels. The performance of the catalysts obtained by means of microwave heating favored the formic acid electro-oxidation with H_2_SO_4_ as the electrolyte. The aerogels’ performance as anodic catalysts was carried out in a microfluidic fuel cell, giving power densities of up to 14 mW cm^−2^ when using mass loads of only 0.1 mg on a 0.019 cm^2^ electrode surface. The power densities of the aerogels obtained by microwave heating gave a performance superior to the resultant aerogel prepared using conventional heating and even better than a commercial Pd/C catalyst.

## 1. Introduction

Noble metals have been widely used as electrocatalysts for energy conversion devices, including fuel cells, water electrolysis, and metal–air batteries [1]. Fuel cells supply energy in a similar way as batteries, although they do not require charging and operate as long as fuel is provided [2,3]. Fuel cells fed with formic acid supply electricity and heat, based on the electrochemical oxidation of fuels in the anode and the reduction of oxygen in the cathode, where H_2_ is formed [4,5]. Pd nanostructures have been used specially in direct formic acid fuel cells that are considered green energy sources for portable electronics and hybrid vehicles due to their high open-circuit voltage, safety and reliability, and low fuel crossover effect, including Pd nanoparticles supported on graphene [6], since nanostructures have simple morphology [7] until hybrid variation of Pd–Cu [8] and Pd–Co [9]. Although these supported materials have great performance towards formic acid oxidation (FAO), the carbonaceous support is still a problem when working with devices that work at high voltages.

Conventional formic acid fuel cells usually employ a physical barrier for the separation of electrodes, which presents many limitations such as membrane fouling and clogging [10]. Therefore, fuel cells working with microfluids would take advantage of laminar flow as a fluid separator and avoid the use of membranes and their drawbacks [11]. Microfluidic fuel cells have the advantage of being portable and of carrying out small-scale processes offering high efficiency of energy conversion. The development of this type of device allows to incorporate the electro-oxidation of formic acid for diverse technological applications on a small scale [12]. On the other hand, miniaturizing the cell may reduce fabrication costs. A study has recently presented a novel microfluidic fuel cell (MFC) that incorporates the innovation of using a laminar flow instead of the conventional solid membrane to separate the fuel and oxidant [13]. Hence, the membrane-related issues are eliminated in this new MFC, which also offers savings in the manufacturing costs. However, it is still necessary to develop effective materials for these types of devices.

The use of catalysts for generating hydrogen from formic acid could minimize the dependence on lithium batteries in a large number of mobile devices [7]. Formic acid can be decomposed catalytically according to the following reactions:HCOOH_(l)_ → H_2 (g)_ + CO_2 (g)_(1)
HCOOH_(l)_ → H_2_O_(l)_ + CO_(g)_
(2)

Noble metals have been extensively studied as catalysts due to their high efficiency, non-toxicity, and stability. Particularly, Pd is widely used in anodes for FAO [9,10]. The synergistic effect of the metallic phase and oxyphilic properties of the Pd surface provides active sites for adsorption and dissociation of formic acid besides providing promoters of oxygen-containing species at low potentials [14,15]. There are a great number of Pd-based catalysts in the bibliography for acid formic oxidation, however, improving their activity is still a requirement in order to be implemented in fuel cells.

Mesoporous materials with low density and a greater number of active sites such as aerogels would allow the use of less mass of the catalyst and, at the same time, to provide a high catalytic activity [12,13,14]. Noble metal aerogels have approximately 90% air and very low contents of the active metal which reduces the cost of the catalysts [15,16]. These materials are commonly obtained by a sol-gel process and supercritical drying [16,17], however, other techniques such as lyophilization allow promising aerogel qualities [18]. Lyophilization, like supercritical drying, shows high efficiency in the formation of metallic aerogels [19,20,21]; this has been demonstrated in works such as Cu(II) cryogels [22] and Pd/CeO_2_-ZrO_2_ alloy aerogels that have been used in the reduction of CO [23] poly(3-sulfopropylmethacrylate) (p(SPM)) cold gels for H_2_ production and mostly organic aerogels as supports for other catalytic materials [24,25,26].

The aerogel synthesis in this work was implemented under microwave radiation. This heating technology allowed, not only the saving of processing time that is usually associated with this type of heating but also to obtain materials with homogeneous and controlled low particles. This fact is quite advantageous, as the particles are usually obtained with more heterogeneous and big particle sizes when the materials are prepared under long conventional heating. The microwave heating allows heating of the bulk precursor solution without gradients, which favors a very good dispersion of nucleation points for the reaction occurring and therefore a better control of the final particle size. Furthermore, the use of microwaves may also influence some chemical reactions and the final products may present some chemical differences in comparison with the ones obtained by conventional heating. In this work, the use of microwave heating provides many benefits such as facile synthesis of homogeneous and low particle size synthesis of Pd aerogels, with high activity towards the oxidation of formic acid in an MFC due to their unique physicochemical characteristics.

## 2. Experimental

### 2.1. Pd Aerogels Chemical Synthesis

The synthesis of Pd aerogels was carried out by adding 10 mL of a 2 mg/mL solution of PdCl_2_ (99%, Sigma-Aldrich ReagentPlus^®^, anhydrous powder, St. Louis, MO, USA) in deionized water into a solution of 240 mg of sodium carbonate (≥99.5%, J.T. Baker^®^) and 40 mg of glyoxylic acid monohydrate (98% Sigma-Aldrich) (ratio 6:1) in 40 mL of deionized water at 67.5 °C (Figure 1) for two hours in all cases. Two different devices were used as a heating source for this initial step of the reaction: conventional heating in a lab oven (CON) and microwave heating (MW). In the case of microwave heating, the reaction temperature was controlled by a thermocouple introduced in the precursor mixture and connected to a proportional–integral–derivative (PID) temperature controller installed in the microwave oven.

Once the first reactions took place, the reduction reaction and gelation process were also carried out by heating the mixture either by conventional heating (CON) or microwave heating (MW) under different operating conditions (i.e., 45 °C or 67 °C and 7 h or 24 h), as described in Table 1. After the gelation process, hydrogels were washed several times using deionized water and ethanol to remove the organic residues in the aqueous solution. Before submitting the clean samples to the lyophilization process, they were frozen with Liquid N_2_, with a volume of 3 mL of deionized water and finally they were dried in a lyophilizer (LYO). A conventional drying in a lab oven (CON) was also performed for comparative proposes.

### 2.2. Physicochemical Characterization

The morphology of the Pd aerogels was characterized using a JEOL JEM-2100F high-resolution transmission electron microscope (HR-TEM) with spherical aberration correction and a scanning electron microscope (SEM, Quanta FEG 650 microscopes from FEI). The crystal structures were measured by X-ray diffraction (XRD; D8-advance diffractometer Bruker) equipped with a CuKα X-ray source (λ = 0.1541 nm, 40 kV, 40 mA), using a step size of 0.02° 2θ and a scan step time of 5 s. The specific surface area and the pore size distribution were determined by nitrogen adsorption–desorption isotherms at −196 °C (Micromeritics ASAP 2020), after an overnight outgassing at 120 °C. The electronic structure of elements was measured by X-ray photoelectron spectroscopy (XPS; K-Alpha+ spectrometer equipped with the Avantage Data System from Thermo Scientific^TM^, Waltham, MA, USA).

### 2.3. Electrochemical Measurements

#### 2.3.1. Electrocatalytic Activity in Half-Cell Configuration

The electrochemical evaluation of the Pd aerogels was carried out in a Biologic VMP3 Potentiostat/Galvanostat using a conventional three-electrode electrochemical cell in acid media at a scan rate of 20 mV s^−1^. A glass-carbon electrode (3 mm) was used as the working electrode, Hg/Hg_2_SO_4_ electrode as the reference electrode, and Pt wire as the counter electrode. The electrocatalyst ink was prepared using each aerogel sample in a mixture of 500 μL of deionized water and 50 μL of Nafion^®^ (5%) per milligram of catalyst. The ink was sonicated for one hour and then 5 μL were deposited over the electrode surface. A similar ink was prepared using commercial Pd/C (20%, Sigma Aldrich, St. Louis, MI, USA) as catalyst and it was used for comparison. The electrolyte was bubbled with N_2_ for 30 min before the electrochemical measurement.

The electrochemical profile for each sample was obtained in cyclic voltammetry (CV) experiments in 0.5 M H_2_SO_4_ within a potential range of 0–1.4 V vs. RHE, where the faradaic processes were visible in a current (i.e., J) that was tested by mg of catalyst.

#### 2.3.2. FAO Performance

The electrocatalytic activity of the Pd aerogels towards FAO were tested by cyclic voltammetry (CV) in a 0.5 M of HCOOH in 0.5 M H_2_SO_4_ electrolyte. As for the electrocatalytic activity in the half-cell configuration, at potential range between 0 and 1.4 V vs. RHE was explored. The results were also compared with the FAO electrocatalytic activity of a commercial reference (Pd/C 20%, Sigma Aldrich).

#### 2.3.3. Stability Performance

The stability performance was carried out by a chronoamperometry (CA) technique at 0.3 V vs. RHE for 24 h at nitrogen atmosphere.

#### 2.3.4. Evaluation of the Microfluidic Fuel Cell System

The description of the MFC used for these experiments has been previously reported [27]. Both the anode and cathode were Pd aerogel samples deposited on Toray carbon paper-060 (TCP) with a transversal area of 0.02 cm^2^. The electrocatalyst loading was 0.1 mg for both electrodes. Linear sweep voltammetry (LSV) was performed by injecting 0.5 M HCOOH with H_2_SO_4_ as the electrolyte in the anode with the evaluated catalyst; and 0.5 M H_2_SO_4_ in the cathode with commercial Pt/C as previous studies reported its best performance [19]. A flow rate of 200 µL min^−1^ was used in the test, i.e., Figure 2.

## 3. Results and Discussion

### 3.1. Physicochemical Characterization

The route selected in this paper, i.e., in situ reduction and subsequent fusion, is one of the two common strategies found in the literature for the preparation of Pd aerogels [28,29,30,31]. This route is essentially a one-pot synthesis that significantly shortens and simplifies the metal nanoparticles aerogels. The aerogels are thus synthetized by mixing noble metal salts (PdCl_2_ in our case) with a strong reducing agent, typically NaBH_4_, LiAlH_4_, hydrazine, sodium citrate, tannic acid or, in our case, glyoxylic acid monohydrate (combined with a base). The two reactants lead to the Pd^2+^ to Pd^0^ reduction at moderate temperatures, typically around 60 °C. Under these conditions, the metal nanoparticles in the sol state tend to aggregate until the stability of the solution is compromised and turned into a gel state (Figure 1). The hydrogel obtained after the Pd aggregation is finally washed several times using deionized water by carefully exchanging the supernatant to ensure the integrity of the formed hydrogel but minimizing the presence of impurities (remnant salts and organic matter) and dried either using supercritical CO_2_ extraction or, in our case, lyophilization.

Glyoxylic acid monohydrate (combined with a base) is a popular reducing agent in the electroless copper plating [32], and it has been used before in the synthesis of metal aerogels [31]. Although the mechanism of the Pd reduction is not fully understood, it is plausible that the contribution of the glyoxylic acid is twofold according to the following equations. First, the disproportionation of the glyoxylic acid in a basic medium occurs:2OCHCOOH + H_2_O → HOCH_2_COOH + HOOC–COOH(3)

The oxalic acid would then react with the PdCl_2_:HOOC–COOH + PdCl_2_ + Na_2_CO_3_ → Pd(COO)_2_ + 2NaCl + H_2_CO_3_
(4)

Finally, the palladium oxalate would be reduced by the glyoxylic acid [32]:Pd(COO)_2_ + OCHCOOH + 2OH^−^ → Pd + 2C_2_O_4_^2−^ + H_2_O + 2H^+^
(5)

In this work, the yield of the Pd aerogel synthesis was 67%.

The XPS analysis was performed to evaluate the composition and electronic structure of the four Pd aerogels obtained. Figure 3 displays their core-level binding energy Pd 3d5/2 (336.8–337.0 eV) and Pd 3d3/2 (340.0–341.0 eV) XP spectra [21,33]. For all samples, each peak can be deconvoluted into two contributions: 335.0 and 340.2 eV for metallic Pd; and 337.0 and 342.4 eV for Pd^2+^ [34]. XPS data revealed that Pd^0^ is the main species on the Pd aerogels surface for PdA-MMT. The high-resolution Pd XPS profiles of samples PdA-CC and PdA-MM are very similar pointing out that the microwave heating produces an analogous reduction process but with a remarkable saving of time (i.e., 7 h vs. 24 h, see Table 1). In addition, it can be also observed that the Pd metallic phase on PdA-MMT aerogel is higher than that on the other samples. This analysis shows that the ratio between Pd^0^ and Pd^2+^ favors Pd^0^ when the aerogels were prepared by microwave heating, and this would probably be one of the main reasons for high stability and performance of that catalyst, as it will be shown below. Therefore, increasing the temperature from 45 °C to 67 °C during the reduction stage would be preferred because it brings about a greater quantity of metallic Pd^0^.

Powder X-ray diffraction (XRD) measurements were performed to evaluate the crystallinity of Pd aerogels samples. Figure 4 illustrates the XRD patterns of the synthesized catalysts. In all XRD patterns, two major diffraction peaks appear at about 40.1° 2θ and 46.6° 2θ, which are ascribed to the (111) and (200) reflection planes of metallic Pd, respectively [35]. These peaks agree with a face-centered cubic crystal structure of Pd (JCPDS# 46-1043) [36]. In addition, the crystallite size was calculated using the Scherrer equation:$$d_{111}=\frac{\;K\lambda }{\beta_{111}\;\cos\;\theta \;}$$

The crystallite sizes for the PdA-CC, PdA-MC, PdA-MM, and PdA-MMT samples were 10.3, 7.2, 6.9, and 7.6 nm, respectively.

The use of microwaves reduces the size of the Pd crystallites, the minimum size being obtained when microwave was used as the heating method in the two stages of the synthesis. The size of the crystal is significantly smaller when using microwaves as a heating method because this heating process is volumetric and heat gradients are minimized. Thus, the reaction occurs uniformly in the precursor mixture. This means that under microwave heating, there are multiple crystallization spots in the precursor solution, whereas in conventional heating, the temperature gradient produces less crystallization spots that grow to form larger particles.

The morphologies of the Pd aerogels were characterized by SEM and the images are presented in Figure 5c–f.

The aerogels obtained present a three-dimensional porous network anchored with nanochains which are extremely thin and make the material look like a sponge with a wide pore size distribution in the range of mesopores and macropores, i.e., Figure 5a,b.

All samples present a similar morphology, although sample PdA-MM (i.e., Figure 5c) seems to present the most open 3D structure. Slight differences in morphology between the different metallic Pd aerogels are attributed, again, to the heating technology applied in their synthesis. Thus, a finer distribution of particles appears in the material synthesized and reduced by microwave heating. This can be explained in Figure 6f where it is observed that the distribution of heat within the solution, by the radiation of microwaves, allows the reaction to occur homogeneously. Instead, larger particles are formed by conventional heating due to the fact that the heat distribution begins at the edges and causes the reaction to occur in an inhomogeneous way, resulting in larger particles that decrease the BET surface area in the aerogels.

TEM images at different magnifications show chains of particles surrounding pores of different sizes. The spherical nanoparticles in those chain structures present different lengths depending on the treatment performed during the synthesis of the aerogels. Particle diameters of up to 30 nm for the aerogels were synthesized in the conventional oven, i.e., Figure 6c, whereas particles less than 9 nm in diameter are characteristic of samples synthesized in MW (Figure 6d–f). To understand the effect of heating on the particle size, the normal distribution of the widths was analyzed, the trend of the means for each sample analyzed is observed in the histograms, being 15.3 nm for PdA-CC, 6.2 nm for PdA-MC, and 5 nm for PdA-MM and Pd-MMT. Therefore, a smaller particle size is attributed to the effect of microwave heating.

According to the different structures observed by SEM and TEM, different porosity and therefore availability of reactive surface area of the aerogels studied was expected, which could be relevant for their further use in electrochemistry. Porous properties of samples were investigated by nitrogen adsorption–desorption isotherm at −196 °C (see Figure 5a). The isotherms of the aerogels are of type II according to the IUPAC classification, which are characteristic of meso–macroporous materials [36] according to the low volume of adsorption at low relative pressures, the sharp increase in the adsorption at high relative pressures, and the absence of a hysteresis loop. Furthermore, the pore size distribution reveals a high volume of mesopores (i.e., 2–50 nm) and macropores (i.e., >50 nm) as can be seen in Figure 5b. The surface area of the samples was determined using the BET equation, giving relatively low values due to the lack of microporosity (i.e., pores < 2 nm) in these samples: 45, 65, 75, and 77 m^2^g^−1^ for PdA-CC, PdA-MC, PdA-MM, and PdA-MMT, respectively. Nevertheless, a trend to increase the BET surface area is observed if microwave heating is used in the different steps during the synthesis.

### 3.2. Electrocatalytic Performances

The electrocatalytic activity for aerogels samples were evaluated using cyclic voltammetry (CV). First, electrochemical profiles were obtained in a 0.5 M H_2_SO_4_ aqueous solution at ambient conditions with a sweep rate of 20 mVs^−1^ (Figure 7a). The peaks detected are attributed to (i) the hydrogen desorption in the 0.1–0.25 V range, (ii) hydrogen adsorption at 0.23 V, (iii) reduction of Pd (II) oxide at 0.65–0.75 V, and (iv) formation of Pd (II) oxide at 1–1.2 V. All these phenomena are present in the cyclic voltammograms of all Pd aerogels. However, the use of microwave radiation during any of the synthesis steps clearly improves the electrochemical activity of the materials.

To quantify this improvement, the electrochemical active surface area (*ECSA*) was evaluated on the electrode surface of each catalyst. The values of *ECSA* for the samples studied in this work were estimated from the cyclic voltammograms (i.e., Figure 7a) by using the reduction charge of Pd (II) oxide according to the following Equation (2):$$ECSA=\frac{Q_{m}}{\;m_{Pd}ed_{m}}$$
where, *Q_m_* denotes coulombic charge (*Q* per µCcm^−2^) for the reduction of Pd (II) oxide achieved by integrating the charges related to the reduction of Pd (II) oxide for the different samples; *m_Pd_* is the mass amount of Pd loaded (g cm^−2^) on the GC electrode surface and *ed_m_* is a constant (424 µC cm^−2^), which corresponds to the reduction of a Pd (II) oxide monolayer [37].

The *ECSA* values of the Pd aerogels samples depicted in Figure 7a are shown in Table 2. *ECSA* values for PdA-CON and the commercial Pd/C catalyst are also included for comparative purposes. As expected from the cyclic voltammograms, there is a great increase in the electrochemical active surface area for the samples prepared using MW. The lower particle size detected by TEM, the lower size of the crystals detected by XRD, and the higher content of the Pd^0^ evaluated by XPS, in samples obtained using microwave heating show clearly that this process has a huge impact on the electrochemical performance of the resulting Pd aerogels. In other words, by means of microwave heating instead of conventional heating for the synthesis of Pd aerogels, the innovation presented in this work, not only the is the processing time reduced but also the electrochemical behavior of these materials is notably enhanced.

This is further corroborated when performing the CV experiments in an electrolyte containing formic acid (i.e., Figure 7b). The evaluation of FAO was carried out in the same range of potential as the CV tests (0.0–1.4 V vs. RHE, see Figure 7a). Comparison of the electrochemical profiles with FAO curves recorded by GC electrodes clearly demonstrated that Pd aerogels offered strong peaks for the electro-oxidation at room temperature conditions [38]. The maximum current values during FAO occur at 0.4 V vs. RHE. Again, the effect of using microwave heating for the synthesis of the Pd aerogels clearly increases their activity in the electro-oxidation of formic acid. Thus, a maximum mass current (J) of 1750 mA mg^−1^ was for the PdA-MMT sample; in the case of PdA-MM and PdA-MC materials, values of 1200 and 1190 mA mg^−1^ were respectively obtained, being the lowest performance for a PdA-CC sample with 700 mA mg^−1^. On the other hand, it seems clear that increasing the temperature of the reduction step from 45 °C to 67 °C makes a difference in the formic acid oxidation activity (see Figure 7b).

As for the formic acid electro-oxidation mechanism with these Pd aerogels, the reaction occurs following two parallel paths, one giving rise to CO_2_ at reasonably low overpotentials through the so-called active intermediate and a chemical dehydration step leading to adsorbed CO, which will be oxidized to CO_2_ at higher potentials [39]. The peaks between 0.2 and 0.6 V for all Pd catalysts (Figure 7d) represent a direct pathway (HCOOH → COOHads/HCOOads + H^+^ + e^−^ → CO_2_ + 2H^+^ + 2e^−^), while the peaks ranging from 0.7 to 0.9 V represent an indirect pathway (HCOOH → COads + H_2_O → CO_2_ + 2H^+^ + 2e^−^) [40]. Since the maximum electrochemical activity of all Pd aerogels studied was measured at ca. 0.4 V, the direct pathway is clearly favored in this case.

The best sample, PdA-MMT, obtained in this work was also compared with the Pd/C commercial catalyst (20 wt%) and PdA-CON in order to show the benefits of using the novel synthesis presented in this work (i.e., microwave heating during the synthesis and lyophilization as the drying procedure). To analyze the catalytic activity of PdA-MMT versus Pd/C, the maximum current intensity shown in Figure 7e should be compared. It can be seen that a current density near to 1900 mA mg^−1^ was observed for the aerogel against almost 300 mA mg^−1^ for Pd/C. Whilst in the case of the behavior of the PdA-MMT vs. the PdA-CON sample, a significant increase in electrochemical activity was detected for the PdA-MMT sample (see Figure 7e).

In order to characterize not only the activity but also the stability of the PdA-MMT sample, a test was carried out for 24 h on this sample and the commercial catalyst Pd/C (Figure 7c,d). The results reveal that the great activity of the sample obtained in this work (PdA-MMT) is totally maintained and stable with time.

### 3.3. Microfluidic Fuel Cell Performance

A microfluidic fuel cell was used for testing samples to verify their activity under real operating conditions, using 0.5 M formic acid with 0.5 M H_2_SO_4_ as the electrolyte for the anodic reaction. Three materials have been tested, namely, PdA-MMT, PdA-CON, and the commercial Pd/C catalysts used as reference materials. Linear sweep voltammetry is a specific electrochemical protocol to discriminate the catalytic activity of the anode materials. The microcell works by pumping 200 µL min^−1^ of the electrolyte fuel (0.5 M HCOOH in 0.5 M H_2_SO_4_) into the anode, and 200 µL min^−1^ of only 0.5 M H_2_SO_4_ into the Pt/C cathode. Both electrodes were normalized to a mass charge of 0.1 mg of each material over an area of 0.019 cm^2^. The polarization curves were obtained for stable open circuit voltage (OCV) and values registered for the PdA-MMT, Pd/C, and PdA-CON samples were 0.88, 0.9, and 0.84 V, respectively. The power density obtained in Figure 7f shows favorable results for the aerogel obtained by MW heating. Small bubbles could be detected on the anode during the experiment, corresponding to the generation of hydrogen. The mass current of the PdA-MMT aerogel is almost three times higher than the commercial Pd/C catalyst (i.e., Figure 7e), corroborating, again, the superior performance of the aerogels obtained by using microwave heating during the synthesis. The superior catalytic activity of PdA-MMT may be attributed to the most active Pd surface. The use of microwave heating during the synthesis of the aerogels leads not only to a better morphology of the aerogels but also to a higher content of metallic Pd^0^. Previous studies reveal that Pd^2+^ species are catalytically inactive for FAO [41]. Therefore, the use of microwave heating has a determinant influence on the production of effective Pd aerogel catalysts.

Table 3 shows the performance of other microfluidic devices from the bibliography in comparison with the one used in this work with the Pd catalyst obtained by the innovative method presented. It may be observed that the configuration used in this work using PdA-MMT as anodic material allows to obtain extremely higher yield per mass of catalyst in terms of density power.

## 4. Conclusions

The use of microwave heating during the synthesis, followed by lyophilization as the drying step, renders Pd aerogels with a particle size smaller than 5 nm in anchored chains. Furthermore, Pd^0^ species are generated in a greater proportion when microwave heating is used during the synthesis. These facts demonstrate that the sample obtained by microwave heating presents a much higher electrochemical active surface area, thus favoring the formic acid electro-oxidation reaction. The PdA-MMT sample showed a three-times superior performance in a microfluidic fuel cell than a commercial Pd/C catalyst, reaching current densities of up to 118.3 mA cm^−2^.

This result shows that the innovative synthesis route presented in this work, using microwave heating for the synthesis process, leads to the production of very competitive aerogels to be used as electrocatalysts.

## Figures and Tables

**Figure 1 materials-15-01422-f001:**
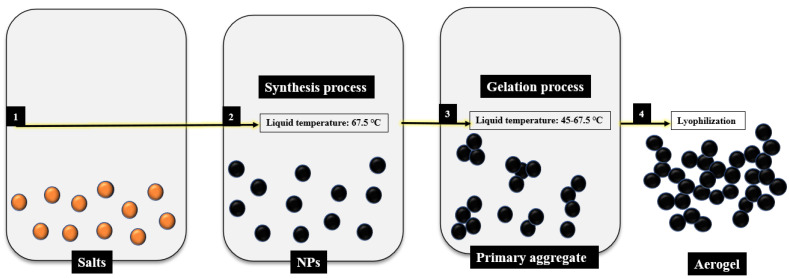
Procedure for obtaining aerogels using the sol-gel methodology in this work.

**Figure 2 materials-15-01422-f002:**
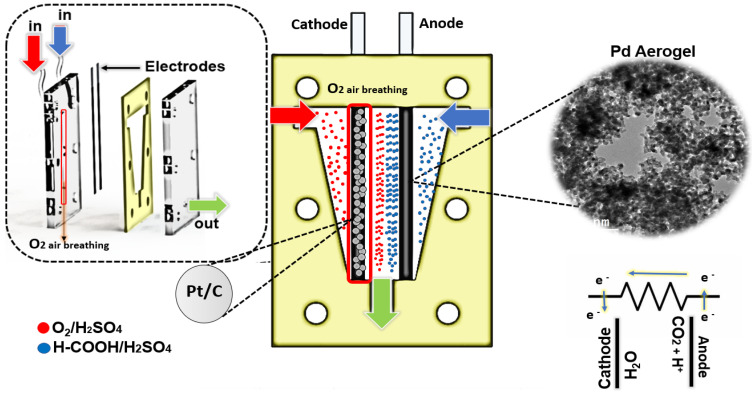
Microfluidic system for formic acid electro-oxidation.

**Figure 3 materials-15-01422-f003:**
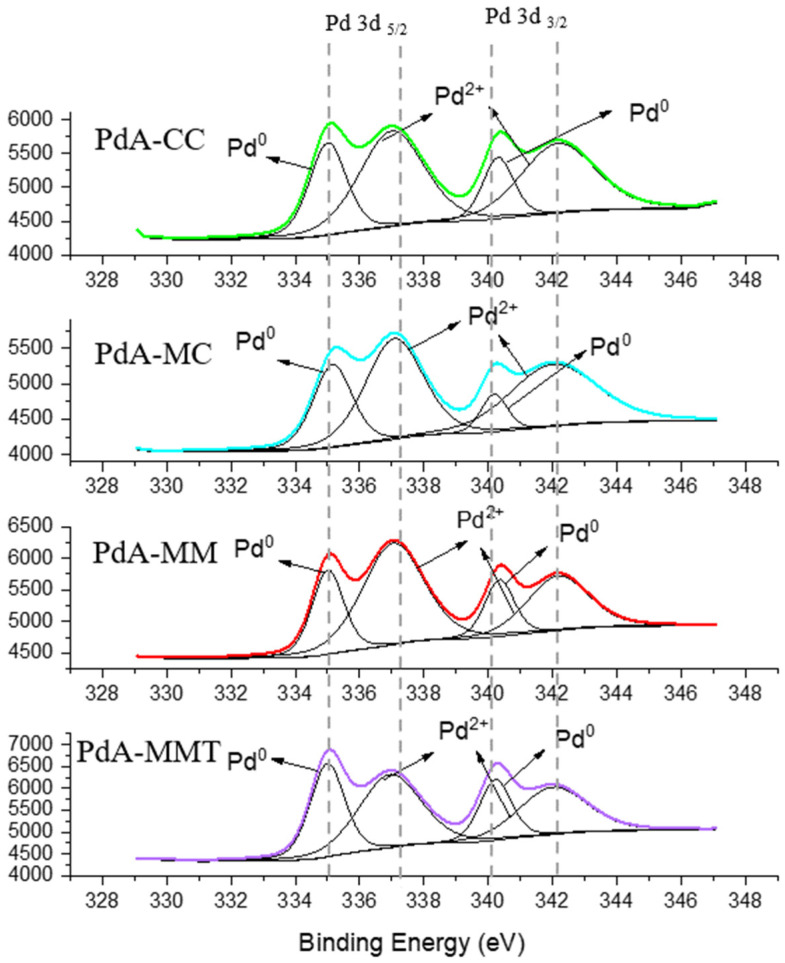
XPS of the Pd aerogels obtained.

**Figure 4 materials-15-01422-f004:**
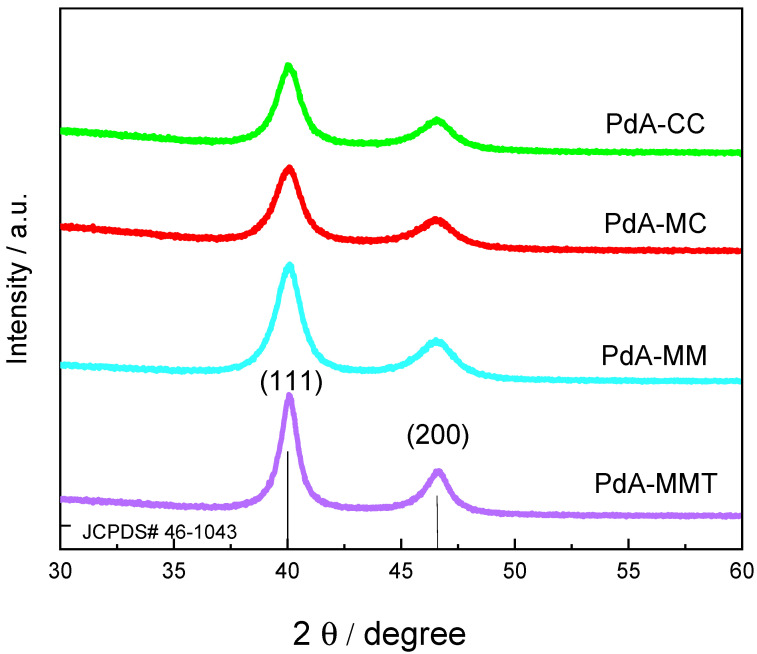
XRD of Pd aerogels samples.

**Figure 5 materials-15-01422-f005:**
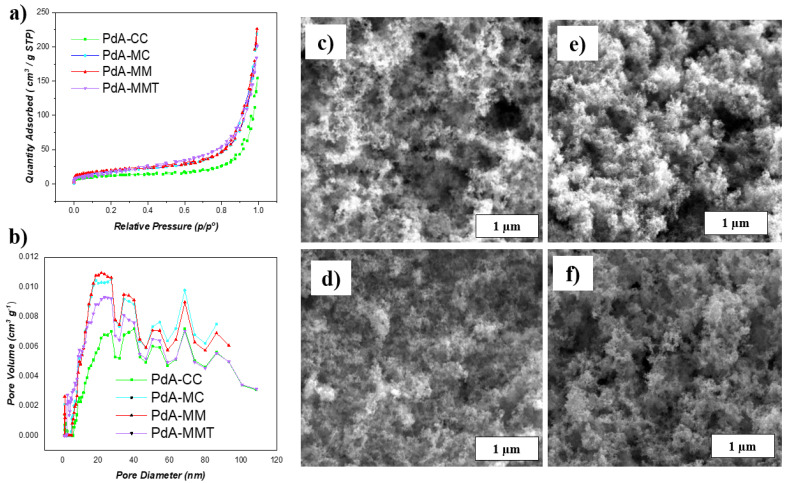
Physicochemical analysis. BET analysis of Pd aerogels: (**a**) N_2_ adsorption isotherms; and (**b**) pore size distribution. SEM of Pd aerogels: (**c**) PdA-CC; (**d**) PdA-MC; (**e**) PdA-MM; and (**f**) PdA-MMT.

**Figure 6 materials-15-01422-f006:**
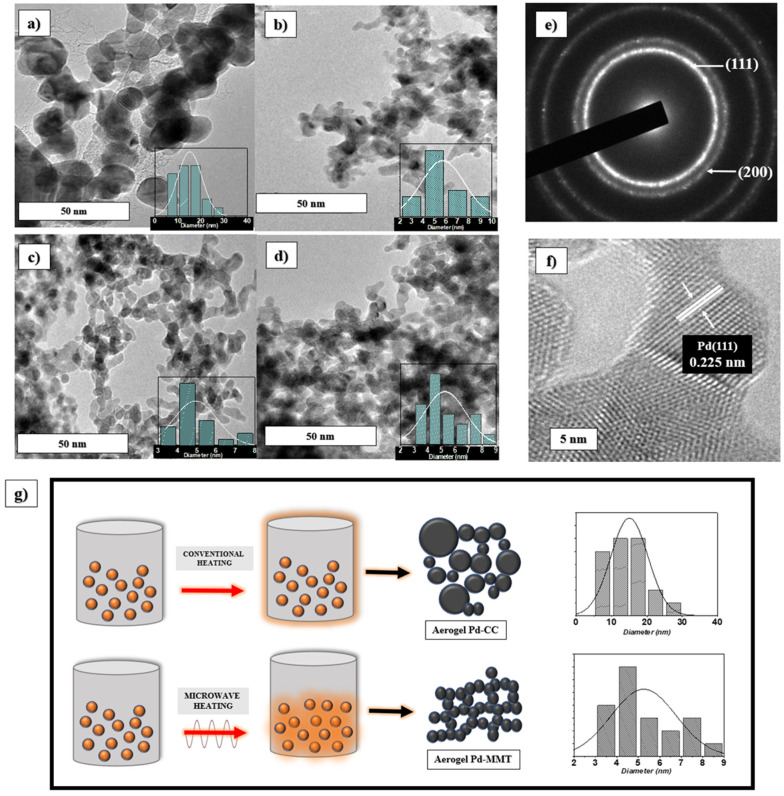
TEM micrographs of Pd aerogels. (**a**) PdA-CC, (**b**) PdA-MC, (**c**) PdA-MM, (**d**) PdA-MMT, (**e**,**f**) crystallographic patterns observed in PdA-MMT aerogel, and (**g**) heating effect on the particle size of Pd-CC and Pd-MMT aerogels.

**Figure 7 materials-15-01422-f007:**
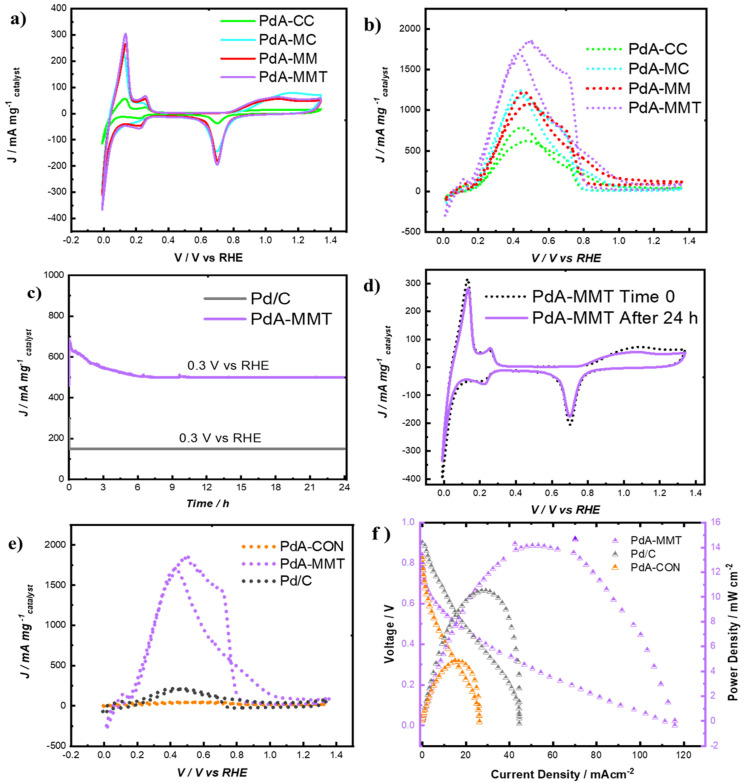
Electrochemical performance. (**a**) Pd aerogels profiles in 0.5 M H_2_SO_4_; (**b**) comparison between the aerogels in FAO; (**c**) stability performance on Pd-MMT vs. Pd/C; (**d**) PdA-MMT CV before and after the CA for 24 h; (**e**) FAO performance between PdA-CON, PdA-MMT, and Pd/C; and (**f**) MFC performance for the best aerogels obtained (PdA-MMT), a commercial catalyst (Pd/C) and an in-lab catalyst obtained by conventional procedure (PdA-CON).

**Table 1 materials-15-01422-t001:** Synthesis procedures and operating conditions to obtain the aerogels of this work.

Sample	Synthesis at 67.5 °C for 2 h	Heating Reduction Device	Reduction Conditions	Drying Device
PdA-CC	CON	CON	24 h/45 °C	LYO
PdA-MC	MW	CON	24 h/45 °C	LYO
PdA-MM	MW	MW	7 h/45 °C	LYO
PdA-MMT	MW	MW	7 h/67 °C	LYO
PdA-CON	CON	CON	24 h/45 °C	CON

**Table 2 materials-15-01422-t002:** Electrochemical active surface area (*ECSA*) values for the samples studied.

Sample	*ECSA* (m^2^/g)
PdA-CON	1.3
PdA-CC	22.1
PdA_MC	22.1
PdA-MM	22.8
PdA-MMT	28.5
Pd/C	16

**Table 3 materials-15-01422-t003:** Anodic catalyst comparison.

Anodic Catalyst	Formic Acid Concentration	Mass Loading/mg cm^−2^	OCV/V	J/mA cm^−2^	W Max/mW cm^2^	Reference
Pd/C	0.5 M	0.7	0.9	7.4	2.9	[42]
Pd/C	0.5 M	0.1	0.9	43	10.5	This work
Pd/MWCNT	0.5 M	1.3	0.9	9.1	2.3	[43]
Pd50Co50/MWCNT	0.5 M	1.2	0.9	5.9	1.75	[38]
PdA-MMT	0.5 M	0.1	0.88	118.3	14	This work
Pd/graphene	0.5 M	2	0.7	30	15.2	[26]
Pt/CN_x_	0.5 M	1	1.1	9.79	3.43	[44]
Pt-Ru	3 M	3	0.47	1.2	12.5	[45]
Au-Pt	1 M	-	1.2	28	12	[46]
Pd	0.5 M	10	0.95	125	26	[47]
Pt	0.5 M	-	1.1	8	2.2	[48]

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
