# Peer review of "Facile Synthesis of Unsupported Pd Aerogel for High Performance Formic Acid Microfluidic Fuel Cell"

_materials, 2022, doi:10.3390/ma15041422_

Round 1

Reviewer 1 Report

L.G. Arriaga et. al. used microwave method to synthesize Pd aerogels, and studied formic acid microfluidic fuel cell. In view of synthesis and fuel cell performance, the overall quality of this paper is qualified for publication after minor modification.

  1. Pd nanomaterials prepared by microwave method should be introduced in Introduction. And the superiority of microwave method to conventional method needs more discussions. For instance, the microwave heating rate, the controllable size for nanomaterials, etc.
  2. Experimental details of the purification of Pd aerogel is missing in 2.1 section.
  3. Raw data in Figure 3 should be provided for better evaluating the deconvolution.
  4. Insets in Figure 6a to 6d need a caption. And the legend name is suggested to be corrected as “diameter (nm)”.
  5. Scatter-line is recommended for graph f in Figure 7.
  6. More anodic catalysts should be included and compared in table 3. Especially papers in the past three years.

Author Response

Thank you for your constructive comments and your thorough revision. Following you will find our comments for every observation you pointed out.

Reviewer 2 Report

The reviewed manuscript entitled: “Innovative Synthesis of Un-supported Pd Aerogel for High Performance Formic Acid Microfluidic Fuel Cell” deals with newly microwave assisted synthesized un-supported Pd NPs aerogel substrate and its use in dilute formic acid electrooxidation microfluidic fuel cell device. The manuscript seems has its research contributive impact though several critical/explanatory remarks and text improving points arise as following:

The title starts with the definition “Innovative synthesis” which by my opinion is somehow misleading and better use “facile synthesis” as the innovative by some wider definition means new or improved synthetic approach which leads to method/product concurrent commercialization on the market and this obviously is not the case.

Introduction part: The author’s state: “In this work, Pd aerogels are synthetized by means of a totally innovative methodology based in microwaved heating during the sol-gel process, followed by lyophilization drying.” is far from being correct as for ex. Herman et al. paper: Multimetallic Aerogels by Template-Free Self-Assembly of Au, Ag, Pt, and Pd Nanoparticles; Chem. Mater. 2014, 26, 2, 1074–1083. The lyophilization of metallic NPs colloidal sols and gels also seems being not something “totally new” for science and the Introduction part should be well-reinforce with critical citation literature on this topic in order the novelty of the current research to self-explant. Also, literature of substrate-supported (carbon, graphene, GO etc) Pd NPs aerogels by lyophilization to be included.

Experimental part: This part is presented unclearly and need serious correction/improvement.

PdCl2 – is this anhydrous or hydrate form? The concentrations of sodium carbonate and glyoxylic acid must be given in mole or weight concrete weight ratios. Also, sodium carbonate and glyoxylic acid instantly/ in situ react each other, formic sodium salt of glyoxylic acid or sodium glyoxylate and CO2 released. So, seems you are adding the Pd precursor salt solution in pre-formed sodium glyoxylate solution. Very important is to give reaction mechanism of PdCl2 chemical reduction by the sodium glyoxylate as reduction agent, its disproportionation etc. Is palladium glyoxylate intermediate? How to explain its decarbonization/reduction at such low (exactly 67.5C) temperature in aqueous media?

What is the practical yield of reduced zero-valent Pd NPs? During the “gelation process” how and why is avoided the primary PdNPs aggregates not to settle out of solution without some viscosity/polymer additive? It seems the authors do not further purify/concentrate the Pd gel stock solution from the remnant salts and organic matter which seems have its role in non-supported aerogel formation during the freeze-drying process. And finally, the obtained final aerogel should be quite brittle, please give much explanation for its manipulation issue before and during catalyst layer preparation! Is it applied in native or grinded/powder form?

Table 1: Heating reduction device assignation is unclear as in the conventional drying oven usually heating and not heat reduction is applied. Please give information for the samples pre-freezing conditions.

Evaluation of the microfluidic fuel cell system.

 Here: “The electrocatalyst loading was 0.1 mg for both electrodes. Linear Sweep Voltammetry (LSV) was performed by injecting 0.5 M HCOOH with H2SO4 as electrolyte in the anode with the evaluated catalyst; and 0.5 M H2SO4 in the cathode with commercial Pt/C as previous studies reported its best performance [20].” It is good you use commercial Pt/C as cathode catalyst but have you performed experiments with both cathode and anode aerogel as catalysts?

Figure 2. Microfluidic System for formic acid. Correct: Microfluidic system for formic acid electrooxidation or something like..

Author Response

(The authors gave the same response as above.)

Reviewer 3 Report

The authors present an innovative method of obtaining Pd aerogel with applications in high performance formic acid microfluidic fuel cell.  
The paper is original, well outlined, it follows the novelties in the field, as demonstrated by the recent bibliography used to support the statements in the paper.
One observation would be related to the absence of arrows in reactions (1) and (2).
The conclusions are supported by experimental data, discussed in a clear way.
From my point of view, the paper can be considered for publication.

Author Response

(The authors gave the same response as above.)

Round 2

Reviewer 2 Report

The manuscript is skillfully revised and the scientific rigor of the manuscript significantly improved. I would endorse the authors to continue their promising research in the field.